# Leigh Syndrome Spectrum: A Portuguese Population Cohort in an Evolutionary Genetic Era

**DOI:** 10.3390/genes14081536

**Published:** 2023-07-27

**Authors:** Manuela Schubert Baldo, Célia Nogueira, Cristina Pereira, Patrícia Janeiro, Sara Ferreira, Charles M. Lourenço, Anabela Bandeira, Esmeralda Martins, Marina Magalhães, Esmeralda Rodrigues, Helena Santos, Ana Cristina Ferreira, Laura Vilarinho

**Affiliations:** 1Research and Development Unit, Human Genetics Department, National Institute of Health Doutor Ricardo Jorge, 4000-055 Porto, Portugal; manuela.baldo@insa.min-saude.pt (M.S.B.);; 2Neonatal Screening, Metabolism and Genetics Unit, Human Genetics Department, National Institute of Health Doutor Ricardo Jorge, 4000-055 Porto, Portugal; 3Inherited Metabolic Disease Reference Center, Lisbon North University Hospital Center (CHULN), EPE, 1649-028 Lisbon, Portugal; 4Inherited Metabolic Disease Reference Center, Pediatric Hospital, Hospital and University Center of Coimbra, 3004-561 Coimbra, Portugal; 5Neurogenetics Department, Faculdade de Medicina de São Jose do Rio Preto, São Jose do Rio Preto 15090-000, Brazil; 6Oporto Hospital Centre, University of Porto, 4099-001 Porto, Portugal; 7Unit for Multidisciplinary Research in Biomedicine, Instituto de Ciências Biomédicas Abel Salazar, Porto University, 4050-313 Porto, Portugal; 8Department of Neurology Porto Hospital and University Centre, EPE, 4050-011 Porto, Portugal; 9Reference Center for Inherited Metabolic Disorders, University Hospital Centre S. João, 4200-319 Porto, Portugal; 10Department of Pediatrics, Hospital Centre, EPE, 4434-502 Vila Nova de Gaia, Portugal; 11Department of Pediatrics, Hospital D. Estefânia, 1169-045 Lisbon, Portugal

**Keywords:** leigh syndrome, mitochondrial disorders, mutational spectrum, clinical spectrum, NGS

## Abstract

Mitochondrial diseases are the most common inherited inborn error of metabolism resulting in deficient ATP generation, due to failure in homeostasis and proper bioenergetics. The most frequent mitochondrial disease manifestation in children is Leigh syndrome (LS), encompassing clinical, neuroradiological, biochemical, and molecular features. It typically affects infants but occurs anytime in life. Considering recent updates, LS clinical presentation has been stretched, and is now named LS spectrum (LSS), including classical LS and Leigh-like presentations. Apart from clinical diagnosis challenges, the molecular characterization also progressed from Sanger techniques to NGS (next-generation sequencing), encompassing analysis of nuclear (nDNA) and mitochondrial DNA (mtDNA). This upgrade resumed steps and favored diagnosis. Hereby, our paper presents molecular and clinical data on a Portuguese cohort of 40 positive cases of LSS. A total of 28 patients presented mutation in mtDNA and 12 in nDNA, with novel mutations identified in a heterogeneous group of genes. The present results contribute to the better knowledge of the molecular basis of LS and expand the clinical spectrum associated with this syndrome.

## 1. Introduction

Mitochondrial disorders (MD) are the most common group of inherited metabolic disorders. These disorders might result from mutations in mtDNA, nDNA, or even genome interplay misfunctioning. In pediatric populations, LS is the most frequent neurodegenerative MD, reaching up to 1:40.000 live births [1,2]. It encloses progressively fatal neurologic failure, high lactate levels, and characteristic symmetrical lesions in brain stem and basal ganglia [2,3]. The diagnostic criteria include clinical, neuroradiological, biochemical, and molecular findings. The developments in molecular diagnosis enabled an easy recognizing of this pathology but also expanded the phenotypical features, as well as the knowledge on the molecular basis of this syndrome. Since Holt et al. first reported m.8993T>G in 1990 [4], several mutations in *MT-ATP6* were subsequently published, also associated with LS, such as 9176T>C [5] and m.10191T>C [6]. The quick advance in nDNA sequencing provided helpful information since biochemical studies and mtDNA sequencing do not allow diagnosis confirmation in every cases. Fatal early-onset presentations, classically seen associated with complex I (CI) deficiency [7,8], were importantly explored through sequencing of structural mitochondrial respiratory chain (MRC) components coding genes. Additional clinical presentations that did not fulfill the clinical and/or radiological criteria for LS but strongly resembled the syndrome were named Leigh-like syndrome (LLS), considering that the evolution of symptoms could be sufficient to complete LS criteria in the future [1,2,9]. Since LLS is diagnosed in patients with nuclear-gene-related mutations, the condition is now denominated Leigh syndrome spectrum (LSS) [9], where LS and LLS are considered a probable continuum of the same disorder [9,10] (Table 1). The molecular advances through NGS and genomics made it possible to obtain huge amounts of information in a timely way, enabling an increase in diagnosis sensitivity, but bringing some uncertainties [11]. The plethora of information produced by these technologies demanded new data for comparison that is not widely available or extensively established. Furthermore, these techniques demand time and expertise to build robustness [12,13]. In mtDNA, for example, identifying a pathogenic variant may allow a condition to be named, but might fail to explain different phenotypes associated with a common underlying genetic lesion and intrafamilial phenotypic differences [14]. To address these questions, precision medicine was evolved through approaches known by “omics” (metabolomics, proteomics, and transcriptomics). The integrative medicine studies dive into deep microenvironments to solve these questions. Despite recent advances it is still too early to apply it in routine testing [15]. While precision medicine is not routine, molecular biology refinements are eventful in promoting faster diagnosis and supporting interventions and counseling to LSS patients.

Unfortunately, there are no curative options for MD yet. Supporting treatment is still the best and only option, and includes dietary management, cofactors administration, specific support of disabilities, and respiratory intervention. Clinical trials for treatments, such as mTOR inhibitors, EPI-743, are undergoing to evaluate its effectiveness in preventing mortality and morbidity, with promising results [16]. In this paper we present data from a cohort of 40 patients diagnosed with LSS highlighting the genetic heterogeneity identified in our cohort, contributing to expand the LSS.

## 2. Materials and Methods

Patients:

A cohort of 40 confirmed LSS individuals, from birth (0 months old (m.o.)) to 36 years, 20 males and 20 females, and respective positive family members, were followed in different clinical reference centers from Portugal. From this group, 11 had already been reported by our group and the remaining 29 were molecularly characterized and reported here for the first time. This study was approved by the Ethics Committees of the National Institute of Health Doutor Ricardo Jorge, and informed consent for genetic studies was obtained from all investigated subjects, or their relatives, in agreement with the Declaration of Helsinki.

DNA extraction, sequencing, and variant analysis:

High-purity DNA was extracted from peripheral blood leucocytes and/or muscle biopsy material according to standard procedures. A custom-made targeted nuclear gene panel was designed using Suredesign software (Agilent 2023 7.10.1.1, Agilent Technologies, Santa Clara, CA, USA) and includes 209 nuclear genes, from which 188 were known to be involved in MD and 21 were candidates based on their role or participation in different pathways that include genes responsible for clinical presentations mimicking MD (Appendix A). Genes of interest were captured with a SureSelect QXT kit (Agilent Technologies), followed by sequencing on the Illumina MiSeq platform. Variant calling and annotation were performed using available commercial programs such as Surecall (Agilent Technologies) and wAnnovar (wannovar.wglab.org/ (accessed on 23 January 2023)). Variants were filtered, taking into account (i) the type of pathogenic variant (missense, frameshift, stop-gain or stop-loss, and splice-site variants), (ii) in silico predictors (SIFT, PolyPhen-2, MutationTaster) [17,18,19] and presence in databases (dbSNP, 1000 Genomes, HGMD professional, ClinVar, ExAC, OMIM, gnomAD), and (iii) the population frequency (variants with a minor allele frequency (MAF) > 1% in the 1000 Genomes Project (http://www.1000genomes.org (accessed on 23 January 2023) and Exome Variant Server databases (http://evs.gs.washington.edu (accessed on 23 January 2023) were filtered out). Furthermore, we searched copy number variations through the analysis of coverage data, using cnMOPS [20]. The entire human mtDNA was enriched by a single amplicon, using back-to-back primers, by long-range PCR [21]. Indexed paired-end DNA libraries were prepared and an equimolar pool was sequenced, according to the manufacturer’s instructions. FASTQ files were aligned to the mtDNA reference sequence with SeqMan NGen (DNAStar). Variant calling and annotation were performed using the following commercial programs: SeqMan NGen and SeqMan Pro (DNASTAR). Variants were filtered considering (i) the type of pathogenic variant, (ii) the population frequency, (iii) in silico predictors and presence in databases (Mitomap, MitImpact2, HmtVar), and (iv) heteroplasmy > 5%. All variants detected that had the potential to be disease-causing were confirmed by Sanger sequencing using the BigDye Terminator Cycle Sequencing Version 3.1 (Applied Biosystems, Foster City, CA, USA), and analyzed on an ABI 3130XL DNA Analyzer. When DNA from additional family members was available, co-segregation studies were also performed [22].

## 3. Results

### 3.1. Mitochondrial DNA Mutations

A total of 28 out of 40 patients, 14 males and 14 female patients, presented mutations in mtDNA. The *MT-ATP6* gene was the most frequent spot of mutations identified in our cohort, corresponding to 48% of the patients. The previously known m.8993T>G (p.Leu156Arg) and m.8993T>C (p.Leu156Pro) are present in 16 patients: 12 patients T>G and four patients T>C (P1 to P16). P7’s mother presented the same mutation m.8993T>G and same heteroplasmy of her child, but exhibiting the neuropathy, ataxia, and retinitis pigmentosa (NARP) phenotype. Two other mutations were identified in the *MT-ATP6* gene: m.9185T>C (p.Leu220Pro) in two patients (P17 and P18) and the m.9176T>G (p.Leu217Arg) in P19, who had low citrulline levels identified on neonatal screening and confirmed on plasma amino acid analysis by ion exchange chromatography. Other, less frequent mutations were identified in our cohort in other mtDNA genes associated with LSS were m.6547T>C (p.Leu215Pro), m.4142G>T (p.Arg279Leu), m.3243A>G, m.10197G>A (p.Ala47Tre), m.10191T>C (p.Ser45Pro), m.13094T>C (p.Val253Ala), m.13513G>A (p.Asp393Asn), and a single-large-scale deletion of 4977bp (Table 2). In *MT-TL1*, the m.3243A>G mutation was identified in P20. The family, namely, mother and two siblings, underwent investigation and all of them harbored the m.3243A>G mutation, but no symptoms were reported at the time. Two other patients had mutations in *MT-ND3*: P21 had m.10191T>C (p.Ser45Pro) mutation, and in P22 we found m.10197G>A (p.Ala47Pro) in nearly homoplasmic state. In *MT-ND5* we identified mutations in three patients: P23 was positive for m.13094T>C (p.Val253Ala), and P24 and P25 with m.13513G>A (p.Asp393Asn). In *MT-ND1*, we identified the mutation m.4142G>T (p.Arg279Leu) in P26 and *MT-CO1* P27 was found to have m.6547T>C (p.Leu215Pro). P28 was diagnosed in the first decade of life through a Southern-blot analysis attesting a single-large-scale deletion of 4977 bp (m.8470-m.13447) in 61% of heteroplasmy in muscle, the most common deletion in the mtDNA (Table 2).

### 3.2. Nuclear DNA Mutations

A targeted 209 nuclear-gene panel allowed molecular characterization of 12 of the 40 patients, five males and seven females, which presented mutations in *SLC19A3*, *NDUFV2*, *NDUFS8*, *NDUFS1*, *POLG*, *SURF1*, *LRPPRC,* and *HIBCH* (Table 3). In nuclear genes associated with CI, we found mutations in four patients: P29 presented in compound heterozygosity in *NDUFS8*, the previously reported mutations c.196C>T (p.Arg66*) and c.287G>A (p.Arg96His); in P30 we identified a compound heterozygous for *NDUFS1*c.470A>T (p.Lys157Met) and c.1798G>C (p.Glu600Gln), which are not reported in databases. Segregations studies confirmed inheritance, and these variants are class IV according to the American College of Medical Genetics (ACMG) scoring system classification [32]. P31 was homozygous for *NDUFV2* splicing mutation c.120+5_120+8delGT, previously described in the literature, which was also identified by prenatal test in her sibling (P32) [22]. In nuclear genes of CIV, we identified mutations in four patients: P33, P34, P35, and P36. P33 and P34 are siblings with different clinical symptoms, and the investigation by NGS retrieved c.19_35dup17* (p.Ala13Cysfs*65)/c.845_846del (p.Ser282Cysfs*9) mutations in *SURF1*, while P35 had isolated psychiatric symptoms and also presented the mutation c.19_35dup17* (p.Ala13Cysfs*65), in homozygous condition, in *SURF1*. P36 was also studied by NGS and was revealed to be homozygous c.74G>A (p.Arg25His) in *LRPPRC*, previously unreported in the literature, which is a class III according to ACMG scoring system classification [32]. Despite fitting into the phenotype and confirmative segregations studies, it remains as VUS, as we will discuss. Co-factors such as thiamine are also related to LSS. We had two patients investigated in our cohort manifesting symptoms of biotin-responsive basal ganglia disease (BBGD), P37, and P38, which presented homozygous mutations c.74dupT (p.Ser26Leufs*19) and c.980-14A>G, respectively, in *SLC19A3* [33]. The patients P39 and P40 had abnormal metabolic work-up with increased dried blood spot propionylcarnitine (C3) in the acylcarnitine profile (by tandem mass spectrometry MS/MS) and increased erytro-2-methyl-2,3-dihydroxybutyric acid and 3-methyl-glutaconic acid in urinary organic acids profile by gas chromatography/mass spectrometry (GC/MS). The NGS gene panel identified in P39 homozygous mutation c.488G>T (p.Cys 163Phe) and in P40 c.129dupA (p.Gly44Argfs*20)/c.910C>T (p.Pro304Ser) in *HIBCH*, coherent to an organic aciduria due to a disturbance in valine catabolic pathway to be further discussed [34,35]. All identified nuclear mutations in this cohort are shown in Figure 1.

## 4. Discussion

The first group of patients in our cohort were molecularly characterized using targeted molecular approaches, the only available in previous decades, as PCR/RFLP and sanger sequencing of selected gene regions. The complete mtDNA sequencing by NGS was an important achievement that enabled the analysis of the total mtDNA genes, increasing the detection power and allowing the identification of several other mutations already linked to other phenotypes that are now part of the LSS. A total of 48% of the patients from our cohort presented mutations in the *MT-ATP6* gene, which is higher than could be expected based on published results [39]. This result could be the reflection of a particular molecular background of Portuguese LSS patients, or a bias associated with the studied cohort. Most index cases had m.8993T>G/C with mutation loads over 95% in muscle and/or blood and recurrent symptoms such as hypotonia, weakness, neurodevelopmental delay or regression, epilepsy, movement disorders, and deafness. The discovery of variable phenotypes of m.8993T>G/C was important to distinguish LSS from and NARP [40], for which the heteroplasmic rates showed that LSS classically display higher than 85% and NARP up to 85%. An additional explanation for the differences in clinical expression is the highly conserved position in *MT-ATP6* resulting in less tolerance to T>G rather than T>C in 8993-point mutations due to the amino acid change, as T>C results in maintaining nonpolar amino acids while the T > G results in substitution of nonpolar amino acid for a charged one, which could produce an important destabilization in the catalytic site of the ATP synthase where this mutation occurs [4,16,41]. In family members of our cohort, we identified two mothers with 75% of heteroplasmy in m.8993T>G: one yet asymptomatic, the other exhibiting NARP syndrome. P7′s mother with NARP displayed the same heteroplasmic rates, but this was calculated in DNA extracted from blood, while P7′s heteroplasmy was accessed in muscle tissue, making the correlation difficult to perform. Our hypothesis for the observed clinical presentations is based not only on the overall different mutation load but also on the distribution of mutated mtDNA by the different tissues. Plasma amino acids profiles can exhibit changes that point to these mutations, namely, low citrulline levels [42,43]. Even at newborn screening (and after excluding urea cycle disorders), low citrulline is suggestive of *MT-ATP6* mutations 8993T>G/C and m.9176T>C/G [44]. A recent paper [45] tried to correlate these mutations to an assertive metabolic profile phenotype, such as the one we found in P19, with low citrulline levels at newborn screening and m.9176T>G (p.Leu217Arg) in homoplasmy [45]. The study’s results showed that reduced citrulline levels and increased propionylcarnitine (C3) and 3-hydroxy-isovaleryl/2-methyl-butyrylcarnitine (C5-OH) might be associated with *MT-ATP6* mutations [45,46]. P17 was investigated due to psychomotor delay, hypotonia, and microcephaly and harbors m.9185T>C (p.Leu220Pro) in 70% of mtDNA molecules while P18 had an LSS and harbors the same mutation in homoplasmy. This mutation is particularly relevant in LSS presenting with ataxia and late-onset cases where ataxia is the core symptom [47,48], albeit fulminant presentation being known as an early-onset LSS with infantile spasms with this mutation [49]. 

In the tRNA^leu^ *MT-TL1*, the MELAS (mitochondrial myopathy, encephalopathy, lactic acidosis, and stroke-like episodes) mutation m.3243A>G has been also identified in different phenotypes such as LSS, maternally inherited deafness and diabetes (MIDD), and isolated myopathy [50,51,52]. In our cohort, P20 presented m.3243A>G and epilepsy, ataxia, hyperlactacidemia, and learning disabilities, with imaging studies characteristics of LSS, an expected phenotype to this mutation [31]. The m.3243A>G usually has a variable heteroplasmy percentage in different tissues and does not respect the usual threshold for MD, as mentioned by Chinnery et al. and by Pann et al. [50,53]. Another mutated gene associated with LSS is *MT-ND3*, where we also found mutations in our cohort. The m.10191T>C (p.Ser45Pro) may manifest as a remarkable CI deficiency, inducing an early neonatal fulminant phenotype [54] or milder cases of encephalopathy, such as P21 [55], and small fiber neuropathy [56], but configuring dystonia and dysarthria as main symptoms [57]. It can also be associated with a slowly clinical progression from infancy to adulthood [58], leading to the association of mtDNA mutations to “spectrum” nomenclature, which was only acknowledged to nDNA mutations [9]. P22 with the mutation m.10197G>A (p.Ala47Thr) in *MT-ND3* also presented a mild phenotype, similar to P21. Mutations in the *MT-ND5* gene were also identified in our cohort. We found m.13094T>C (p.Val253Ala) in homoplasmy in P23, a mutation previously described associated with cerebellar syndromes [59], CI deficiency, or combined complexes deficiencies [60], presenting our patient a marked hypotonia. In pediatric populations, LSS appears to represent a less favorable prognosis [61]. P24 and P25 presented m.13513G>A (p.Asp393Asn), a mutation that has already been outlined by Ruiter et al. in a patient with CI deficiency, cardiac rhythm perturbation, and optic atrophy [62], with our patients presenting cardiomyopathy and hypotonia, and ataxia and apnea, respectively. Our cohort also includes a patient with m.4142G>T (p.Arg279Leu) in *MT-ND1* (P26 [30]) and another one with m.6547T>C in *MT-CO1* (P27 [30]).

Another group of mutations in mtDNA are the rearrangements, such as single-large-scale mtDNA deletions (LMD), a genetic cause of MD expressing a wide range of clinical features. The rearrangements result into a spectrum of syndromes: Pearson, Kearns Sayre Syndrome (KSS), a subtype of chronic progressive external ophthalmoplegia (CPEO), and CPEO itself [63,64]. P28 was investigated due to epilepsy, short stature, cerebellar ataxia, and tremor, and Southern-blot analysis detected an LMD of 4977bp-del (m.8470-m.13447). This association is extremely rare because the early-onset spectrum of LMD, Pearson syndrome, is usually a potentially fatal disorder disabling the progression of the spectrum to other clinical entities, namely, LSS, which makes the association rare [65,66]. Despite the prognosis, the patients who survive may develop LSS or a multiorgan deficiency in the future [63]. 

The patients of our cohort identified with nDNA mutations were all screened by NGS, which represents a satisfactory middle-term cost-utility approach, providing a significant amount of data in a reasonable time, delivering a broader diagnostic performance when set side by side to conventional procedures. The complexes are the major structural components of the MRC resulting in mitochondrial optimized ATP generation [57]. As we observed in our cohort, the CI and CIV most frequently harbor pathogenic mutations, followed by non-OXPHOS defects. In total, in nDNA we identified ten pathogenic mutations, three likely-pathogenic variants, one reclassified in our perspective through ACMG, symptomatology, and biochemical analysis, and one VUS that did not meet criteria to be reclassified. 

The CI is the first and largest MRC complex [67] and is widely associated with LSS, with several genes involved [3,57], resulting in 33 subtypes of mitochondrial CI deficiency (MCID), from first year of life rapidly progressing to fatal course [68,69] to a wide range of presentations [70,71,72]. P30 presented the c.470A>T (p.Lys157Met)/c.1798G>C (p.Glu600Gln) variants in *NDUFS1*, which are not reported in databases so far. Standard classification scores by ACMG resulted in class IV, likely pathogenic, for both variants. In silico predictors such as Polyphen-2 and SIFT described them as deleterious, and Mutation Taster as probably damaging. Segregation studies showed heterozygous parents. The clinical manifestations are described in LSS and CI deficiency, altogether supporting a probable pathogenicity. P31 displayed, with homozygous mutation, c.120+5_120+8delGT in *NDUFV2*, a classic *NDUFV2* phenotype represented by hypertrophic cardiomyopathy and encephalopathy [73,74], alongside with a rare skin condition. In less frequent cases, *NDUFV2* mutations manifest as LSS, which was reported by Liu et al. in two families [75]. 

The CIV Is also a critical part of the MRC as the terminal catalytic site to transfer electrons into oxygen [76]. *SURF1* is a gene that plays part in the assembly of CIV, while *LRPPRC* impacts the mitochondrial translation [77]. We identified P35 with *SURF1* mutations displaying atypical presentation with isolated psychiatric symptoms represented by visual hallucinations with suggestive LSS neuroimaging. Molecular assessment retrieved homozygous c.19_35dup17* (p.Ala13Cysfs*65) in *SURF1*, a gene typically related to a group of homogeneous symptoms (hypertrichosis [78] and peripheral neuropathy), specific imaging findings (leukodystrophy), and high mortality in childhood [79]. However, heterogeneous phenotypes linked to *SURF1* mutations have been reported [80]. P36 was investigated for encephalopathy and brain atrophy. NGS revealed the homozygous mutation c.74G>A (p.Arg25His) in *LRPPRC* gene, a rare variant in population, classified as ACMG class III. Segregation studies were performed, confirming inheritance in our patient. In silico predictors are controverse; Polyphen-2 and SIFT pointing to deleterious role, and Mutation Taster to benign role, while ClinVar classifies it as a VUS associated with congenital lactic acidosis Saguenay-Lac-Saint-Jean, a subtype of LSS [81]. This feature gives us a likely connection between the genotype and the patient’s phenotype, with this variant having the potential to be, in the future, reclassified as likely pathogenic. 

Thiamine pyrophosphate is an important cofactor of several steps in energy production and SLC19A3, a biotin-dependent thiamine transporter in the central nervous system, is fundamentally present in the basal ganglia system [82]. BBGD is a well-known cause of LSS [83] frequently manifesting in childhood with vomiting, seizures, encephalopathy, movement disorders, and free-thiamine dosage in cerebrospinal fluid typically low to absent [84,85]. P37 and P38 investigated in our cohort had mutations in *SLC19A3*. P37 developed neonatal epileptic encephalopathy with homozygous c.980-14A>G in *SLC19A3* and P38 displayed episodic ataxia with homozygous c.74dupT (p.Ser26Leufs*19). Both patients are under treatment with biotin and thiamine and had important recovery of symptoms, which demonstrates the importance of recognizing the phenotype and starting effective intervention [82,83]. 

Organic acidurias are also a proven cause of LSS. In the case of *ECHS1* and *HIBCH* genes, encoding a hydratase and a hydrolase enzyme, respectively, in the final valine metabolism, LSS is the most common clinical manifestation of these genetic deficiencies [86,87,88]. Frequent symptoms identified in these cases are developmental delay or regression, hypotonia, and epilepsy [89]. A recent Spanish study demonstrated that *HIBCH* deficiency was completely correlated to LSS, and *ECHS1* deficiency displayed LSS, but also fatal neonatal lactic acidosis and paroxysmal dystonia. The biochemical profile was divergent between them, which could guide the diagnosis [90]. P39 was confirmed with HIBCH deficiency, being homozygous for c.488G>T (p.Cys 163Phe). P40 is a compound heterozygous for a known pathogenic mutation in *HIBCH* c.129dupA (p.Gly44Argfs*20) alongside an unreported variant c.910C>T (p.Pro304Ser). The new variant meets ACMG criteria as class III. In silico estimates in Polyphen-2, SIFT, Mutation Taster suggest a deleterious behavior. Also, in this case, the positive segregation pattern, clinical manifestations expressed as LSS and biochemical analysis of acylcarnitine profile displaying elevation of 3-hydroxyisobutyryl carnitine, and urinary organic acids showing intermediate metabolites of valine degradation were associated with a reported pathogenic variant we would consider to be a likely pathogenic variant, ACMG Class IV, under this information.

## 5. Conclusions

The present work presents data on a Portuguese cohort of 40 LS patients and allows the expansion of the molecular knowledge on this syndrome and the clinical spectrum associated with it. The present data stress the high clinical and genetic heterogeneity associated with this syndrome. The evolution of the molecular technics from PCR/RFLP to sanger sequencing and finally NGS was a path leading to an unprecedent increase in sensitivity in the diagnosis of LS. The adoption by our lab of NGS allowed the clarification of unsolved cases for many years, being useful in revealing the molecular etiology in children and adult patients by identifying new variants, characterizing more cases, and expanding the LSS scenery of mutations, since rare and nondescribed variants were pivotal for our cases; albeit this is only the beginning.

## Figures and Tables

**Figure 1 genes-14-01536-f001:**
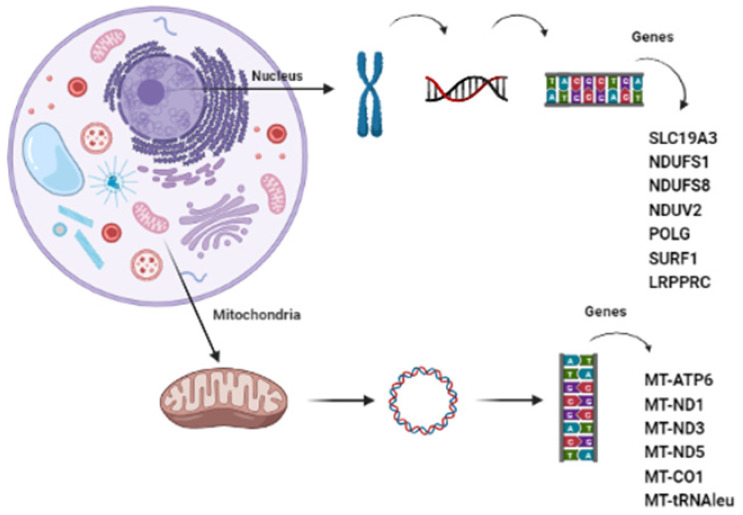
Mitochondrial and nuclear genes with mutations in our cohort, implicated in LSS.

**Table 1 genes-14-01536-t001:** LSS clinical and radiological features [9].

	Leigh Syndrome Spectrum (LSS)
Clinical features *	Metabolic decompensation (elevated lactate levels in blood and/or cerebrospinal fluid) and/or characteristic symptoms during acute illness (episodic). Neurologic manifestations: hypotonia, spasticity, movement disorders (dystonia, chorea), cerebellar ataxia, and peripheral neuropathy. Other systems manifestations: hypertrophic cardiomyopathy, anemia, renal tubulopathy, liver involvement, ptosis, and muscle weakness.
Radiological features	Multiple symmetric bilateral lesions in the basal ganglia, thalamus, brain stem, dentate nuclei, and optic nerves **.

Legend: * Includes LLS features, such as: no evidence of abnormal energy metabolism, atypical neuropathology, and/or incomplete evaluation, to the classic LS criteria resulting in LSS. ** Unilateral lesions in brain stem, basal ganglia, spinal cord, cerebellum; variation in the distribution or character of lesions or with the additional presence of unusual features, such as extensive cortical destruction (neuronal loss).

**Table 2 genes-14-01536-t002:** Results of investigation of LSS patients’ mtDNA in this cohort.

	Gender	Age of Diagnosis	Gene		Mutation Data			Symptoms
				Variant	Heteroplasmy	Status (Mitomap) [22]	Reference	
P1	F	7 y	MT-ATP6	m.8993T>G (p.Leu156Arg)	Muscle: 95%	P	Holt, 1990 [4]	Psychomotor regression, loss of contact, lethargy, generalized seizures, dystonia.
P2	F	2 y	MT-ATP6	m.8993T>G (p.Leu156Arg)	Muscle: homoplasmic	P	Holt, 1990 [4]	Hypotonia, progression to apnea after febrile illness, dystonia.
P3	F	8 y	MT-ATP6	m.8993T>G (p.Leu156Arg)	Blood: homoplasmic	P	Holt, 1990 [4]	Intellectual impairment, spastic paraparesis, hyperlactacidemia.
P4	F	9 y	MT-ATP6	m.8993T>G (p.Leu156Arg)	Blood: 95%	P	Holt, 1990 [4]	Mitochondrial encephalopathy.
P5	M	1 y	MT-ATP6	m.8993T>G (p.Leu156Arg)	Muscle: 95%	P	Holt, 1990 [4]	Hypotonia, myoclonic epilepsy, regression of milestones.
P6	F	29 y	MT-ATP6	m.8993T>G (p.Leu156Arg)	Blood and muscle: >95%	P	Holt, 1990 [4]	LSS
P7	F	16 y	MT-ATP6	m.8993T>G (p.Leu156Arg)	Muscle: 75%	P	Holt, 1990 [4]	LSS
P8	M	6 y	MT-ATP6	m.8993T>G (p.Leu156Arg)	Muscle: 85%	P	Holt, 1990 [4]	LSS
P9	M	1 y	MT-ATP6	m.8993T>G (p.Leu156Arg)	Blood and muscle: homoplasmic	P	Holt, 1990 [4]	LSS
P10	F	1 y	MT-ATP6	m.8993T>G (p.Leu156Arg)	Muscle: 90%	P	Holt, 1990 [4]	Neurodevelopmental delay, hypotonia, epilepsy.
P11	F	19 y	MT-ATP6	m.8993T>G (p.Leu156Arg)	Muscle: 95%	P	Holt, 1990 [4]	Deafness, peripheral neuropathy, ataxia.
P12	F	2 y	MT-ATP6	m.8993T>G (p.Leu156Arg)	Muscle: 98%	P	Holt, 1990 [4]	Failure to thrive, eosinophilic esophagitis.
P13	F	23 y	MT-ATP6	m.8993T>C (p.Leu156Pro)	Muscle: >95%	P	De Vries, 1993 [23]	LSS
P14	M	8 y	MT-ATP6	m.8993T>C (p.Leu156Pro)	Muscle: >95%	P	De Vries, 1993 [23]	LSS
P15	F	10 y	MT-ATP6	m.8993T>C (p.Leu156Pro)	Blood and muscle: >90%	P	De Vries, 1993 [23]	Numbness, ataxia.
P16	M	8 y	MT-ATP6	m.8993T>C (p.Leu156Pro)	Muscle: 98%	P	De Vries, 1993 [23]	Mitochondrial disorder.
P17	F	1 y	MT-ATP6	m.9185T>C (p.Leu220Pro)	Muscle: 70%	P	Moslemi, 2005 [24]	Neurodevelopmental delay, microcephaly and hypotonia.
P18	M	32 y	MT-ATP7	m.9185T>C (p.Leu220Pro)	Blood: homoplasmic	P	Moslemi, 2005 [24]	LSS
P19	M	0 m.o.	MT-ATP6	m.9176T>G (p.Leu217Arg)	Blood: homoplasmic	P	Thyagarajan, 1995 [25]	Still asymptomatic (studied to clarify newborn screening alterations).
P20	M	10 y	MT-TL1	m.3243A>G	Muscle: 73%	P	Poulton, 1988 [26]	Epilepsy, learning disability, ataxia, hyperlactacidemia.
P21	F	4 y	MT-ND3	m.10191T>C (p.Ser45Pro)	Blood: 80%, muscle: 90%	P	Taylor, 2001 [6]	Hypotonia, neurodevelopmental delay, recurrent respiratory infections in first months of life, epilepsy.
P22	M	7 y	MT-ND3	m.10197G>A (p.Ala47Tre)	Muscle: 100%	P	Kirby, 2004 [27]	Epilepsy, failure to thrive, recurrent illness, hypotonia, high serum lactate levels.
P23	M	11 y	MT-ND5	m.13094T>C (p.Val253Ala)	Blood: 100%	P	Valente, 2009 [28]	Severe hypotonia.
P24	F	8 m.o.	MT-ND5	m.13513G>A (p.Asp393Asn)	Blood: 70%	P	Chol, 2003 [29]	Neurodevelopmental delay, cardiomyopathy, hypotonia.
P25	M	7 y	MT-ND5	m.13513G>A (p.Asp393Asn)	Blood: 58%, muscle: 88%	P	Chol, 2003 [29]	Ataxia and apnea.
P26	M	12 y	MT-ND1	m.4142G>T (p.Arg279Leu)	Muscle: 80%	R	Pereira, 2019 [30]	Spastic deambulation with cerebellar signs, hyperreflexia, clonus, dysarthria, dystonia, cognitive impairment.
P27	M	13 y	MT-CO1	m.6547T>C (p.Leu215Pro)	Muscle: 50%	R	Pereira, 2019 [30]	Hypotonia, hyperlactacidemia.
P28	M	7 y	MT-ATP6 to MT-ND5	4977bp-del (m.8470-m.13447)	Muscle: 61%	P	Vilarinho, 1997 [31]	Epilepsy, short stature, cerebellar ataxia, tremor.

Legend: LSS: Leigh syndrome spectrum; y: year; F: female, M: male P: confirmedly pathogenic; R: reported; m.o.: months old.

**Table 3 genes-14-01536-t003:** Results of investigation of LSS patients’ nDNA in this cohort.

Patient	Gender	Age of Diagnosis	Gene	Mutation Data								Symptoms
				Allele 1	Reference	ClinVar	ClinVar Phenotype	Allele 2	Reference	ClinVar	ClinVar Phenotype	
P29	M	36 y	*NDUFS8*	c.196C>T (p. Arg66*)	Nogueira, 2019 [22]	ND	ND	c.287G>A (p.Arg96 His)	Nogueira, 2019 [22]	ND	ND	Short stature, Leigh syndrome compatible neuroimaging.
P30	M	2 y	*NDUFS1*	c.470A>T (p.Lys157 Met)	ND	ND	ND	c.1798G>C (p.Glu600 Gln)	ND	ND	ND	Axial hypotonia, failure to thrive, neurodevelopmental delay, nystagmus.
P31	F	2 y	*NDUFV2*	c.120+5_ 120+8 delGT	Bénit, 2003 [36]	P	Mitochondrial Complex I deficiency	c.120+5_ 120+8 delGT	Bénit, 2003 [36]	P	Mitochondrial Complex I deficiency	Severe eczema, epilepsy, axial hypotonia, neurodevelopmental delay, suspicion of Leigh syndrome.
P32	M	prenatal		c.120+5_ 120+8 delGT	Bénit, 2003 [36]	P	Mitochondrial Complex I deficiency	c.120+5_ 120+8 delGT	Bénit, 2003 [36]	P	Mitochondrial Complex I deficiency	Prenatal screening, P31 sibling.
P33	F	3 y	*SURF1*	c.19_35dup17* (p.Ala13 Cysfs*65)	Tiranti, 1998 [37]	P	LSS	c.845_846 del (p:Ser282 Cysfs*9)	Tiranti, 1998 [37]	P/LP	LSS	Ataxia, hypotonia, tremors, Tetralogy of Fallot.
P34	F	1 y		c.19_35dup17* (p.Ala13 Cysfs*65)	Tiranti, 1998 [37]	P	LSS	c.845_846 del (p:Ser282 Cysfs*9)	Tiranti, 1998 [37]	P/LP	LSS	Neurodevelopmental delay, hypotonia, bradycardia.
P35	M	15 y		c.19_35dup17* (p.Ala13 Cysfs*65)	Tiranti, 1998 [37]	P	LSS	c.19_35dup17* (p.Ala13 Cysfs*65)	Tiranti, 1998 [37]	P	LSS	Visual hallucinations.
P36	F	4 y	*LRPPRC*	c.74G>A (p.Arg25 His)	ND	VUS	Congenit lactic acidosis	c.74G>A (p. Arg25 His)	ND	VUS	Congenit lactic acidosis	Encephalopathy, brain atrophy.
P37	F	1y	*SLC19A3*	c.74dupT (p.Ser26 Leufs*19)	Debs, 2010 [33]	P	BBGD	c.74dupT (p.Ser26 Leufs*19)	Debs, 2010 [33]	P	BBGD	Neonatal epileptic encephalopathy responsive to biotin, thiamine, Coenzyme Q10 and riboflavin.
P38	F	5y		c.980-14A>G	Debs, 2010 [33]	LP	BBGD	c.177G>A (p.Trp59*)	Debs, 2010 [33]	ND	ND	Episodic ataxia responsive to biotin, thiamine, Coenzyme Q10 and riboflavin.
P39	M	3 m.o.	*HIBCH*	c.488G>T (p.Cys 163Phe)	Wojcik, 2019 [38]	LP	HIBCH deficiency	c.488G>T (p.Cys 163Phe)	Wojcik, 2019 [38]	LP	HIBCH deficiency	Stop of progression of the neurodevelopment and feeding difficulties.
P40	F	6 y		c.129dupA (p.Gly44 Argfs*20)	Peters, 2015 [35]	LP	HIBCH deficiency	c.910C>T (p.Pro304 Ser)	ND	ND	ND	Axial hypotonia, psychomotor regression after intercurrent illness, ataxia.

Legend: ND: not described; P: pathogenic; LP: likely pathogenic; m.o.: months old.

## Data Availability

Data is contained within the article.

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
