# Peer review of "Leigh Syndrome Spectrum: A Portuguese Population Cohort in an Evolutionary Genetic Era"

_genes, 2023, doi:10.3390/genes14081536_

Round 1

Reviewer 1 Report

The authors of manuscript entitled “Leigh Syndrome spectrum: a Portuguese population cohort in an evolutionary genetic Era” investigated by NGS the presence of mutations in mtDNA and nuclear DNA genes in 40 Portuguese patients affected by Leigh syndrome spectrum. 28 of 40 patients have mutations on genes located in mitochondrial DNA, both in Heteroplasmy or homoplasmic forms. In particular, the mutations affected 6 mitochondrial genes. In the other 12 patients, mutations were found in seven nuclear genes, which code for mitochondrial proteins. The study associated mutations of mitochondrial and nuclear genes with Leigh Syndrome spectrum, and some of these mutations appear to have been identified for the first time in patients affected by LSS, or mitochondrial diseases.

The study shows the panel of investigated genes is useful and powerful for detecting mutations associated to mtDNA and nuclear DNA in patients with associated-LSS phenotypes. Although mutation analysis has a flaw due to the investigation of a limited number of investigated genes (listed in Appendix A).

Minor revisions

Overall, the discussion is too long and should be shortened, due to repetitive descriptions of mutations found in patients. For example, the authors should only report the overall description of the possible effect of the mutated proteins on each patient’s the metabolism. Furthermore, the authors should better highlight the mutations they identified for the first time in patients with mitochondrial disease.

Line 142: the reference 21 should be changed with reference n.24 Vilarinho, L.; Maia, C.; Coelho, T.; Coutinho, P.; Santorelli, F. M. Heterogeneous Presentation in Leigh Syndrome. 502 J. Inherit. Metab. Dis. 1997, 20 (5), 704–705. https://doi.org/10.1023/A:1005330611147.

Line 219: The authors mentioned “plasma amino acid profiles” changed in patients, but no data or references are given. Specify the sources of this data.

Line 222: the reference to “A recent paper” is missing.

Author Response

Author’s response to reviewer

>Reviewer 1: 

Minor revisions:

Overall, the discussion is too long and should be shortened, due to repetitive descriptions of mutations found in patients. For example, the authors should only report the overall description of the possible effect of the mutated proteins on each patient’s the metabolism. Furthermore, the authors should better highlight the mutations they identified for the first time in patients with mitochondrial disease.

Answer: we have rewritten and shorten the discussion making clear which mutations were identified for the first time. 

Line 142: the reference 21 should be changed with reference n.24 Vilarinho, L.; Maia, C.; Coelho, T.; Coutinho, P.; Santorelli, F. M. Heterogeneous Presentation in Leigh Syndrome. 502 J. Inherit. Metab. Dis. 199720 (5), 704–705. https://doi.org/10.1023/A:1005330611147.

Answer: the change has been made accordingly.

Line 219: The authors mentioned “plasma amino acid profiles” changed in patients, but no data or references are given. Specify the sources of this data.

Answer: the additional data has been added to the text.

Line 222: the reference to “A recent paper” is missing.

Answer: the missing reference has been added to the text: Tise, C. G.; Verscaj, C. P.; Mendelsohn, B. A.; Woods, J.; Lee, C. U.; Enns, G. M.; Stander, Z.; Hall, P. L.; Cowan, T. M.; CusmanoOzog, K. P.  MTATP6  Mitochondrial Disease Identified by Newborn Screening Reveals a Distinct Biochemical Phenotype. Am. J. Med. Genet. A. 2023, 191 (6), 1492–1501. https://doi.org/10.1002/ajmg.a.63159.

Reviewer 2 Report

Baldo et al. analyzed molecular and clinical data on a Portuguese cohort of 40 cases diagnosed with Leigh syndrome spectrum (LSS). The authors have found that 28 patients carried out mutations in the mitochondrial DNA (mtDNA) while the remaining 12 patients in the nuclear genome. The authors also identified new mutations in a heterogeneous group of genes. The study is interesting and clinically very relevant.

However, the results are not always well described and there is a little bit of confusion regarding what is new and what is already known. Material and Methods appear to be incomplete. Literature is poorly cited.

Specific comments are found below:

-       Lines 63-64: Please, provide relative reference.

-    Table 1: To make the Table more comprehensive, I would suggest to the authors to provide the genes affected. Please, include also the relative references.

-       Material and Methods: In the “Patient” section, line 88, I would suggest to put “X” months -36 years old. Please, indicate in the “11” and “29” group how many patients are males and females.

-       Line 96: I would change “Methods” with “DNA extraction, sequencing and variants analysis”.

-   Line 127: I would suggest to call this section “Mitochondrial DNA mutations”

-       Line 128: For the “28” group, would it be possible to include how many patients are males and females? I would include this piece of information (how many males vs females) in all the manuscript when the information is available.

-    The authors reported “metabolic” alterations in some patients. How metabolomic analysis was performed?

-       Table 2: the Legend of this table is misleading. The authors claim that the table “summarizes” the “Results of investigation of LSS patients’ mtDNA in this study, but cite other groups work. I would suggest to clarify it. The authors should clearly indicate what is new and what has been already reported by others. In this regard: 1) more references should be included, and 2) in the manuscript should be specified that the mutations related to MT ATP-6 are found in “only” 10% of Leigh Syndrome patients. There is a very nice work from Marni J. Folk (MT-ATP6 mitochondrial disease variants: phenotypic and biochemical features analysis in 218 published cases and cohort of 14 new cases, Hum Mutat, 2019) that I would recommend to consider and discuss.

-       Table 3: Same comment provided for Table 2.

-       Figure 1: I would simplify the Figure by including only one eukaryotic cell, using two different arrows to indicate the nuclear and the mitochondrial compartment.

-       Line 224: To which study the authors are referring here? Please specify it.

-  Did the authors measured/recorded any metabolic dysfunction and/or immunodeficiencies associated with their mitochondrial variants? In the “Discussion” it seems that this is the case, but the information should be included in the “Result” section.

Minor English editing is required.

Author Response

Author’s response to reviewer

>Reviewer 2:

Comments and Suggestions for Authors

Baldo et al. analyzed molecular and clinical data on a Portuguese cohort of 40 cases diagnosed with Leigh syndrome spectrum (LSS). The authors have found that 28 patients carried out mutations in the mitochondrial DNA (mtDNA) while the remaining 12 patients in the nuclear genome. The authors also identified new mutations in a heterogeneous group of genes. The study is interesting and clinically very relevant.

However, the results are not always well described and there is a little bit of confusion regarding what is new and what is already known. Material and Methods appear to be incomplete. Literature is poorly cited.

 Specific comments are found below:

-  Lines 63-64: Please, provide relative reference.

Answer: we added the reference according to the suggestion: Rahman, S.; Thorburn, D. Nuclear Gene-Encoded Leigh Syndrome Spectrum Overview.

-    Table 1: To make the Table more comprehensive, I would suggest to the authors to provide the genes affected. Please, include also the relative references.

Answer: the changes have been done to address this specific situation including the genes on the tables and citations.

-  Material and Methods: In the “Patient” section, line 88, I would suggest to put “X” months -36 years old. Please, indicate in the “11” and “29” group how many patients are males and females.

Answer: the change has been made according to the request replacing years by “months-old” where the patients were described as “0y.o”. We also indicated on each group, mitochondrial and nuclear, how many males and females are included in each group.

-  Line 96: I would change “Methods” with “DNA extraction, sequencing and variants analysis”.

Answer: the change has been made according to the request.

-  Line 127: I would suggest to call this section “Mitochondrial DNA mutations”

Answer: the change has been made according to the request.

-       Line 128: For the “28” group, would it be possible to include how many patients are males and females? I would include this piece of information (how many males vs females) in all the manuscript when the information is available.

Answer: the information was inserted according to the suggestion, especially in the “materials and methods” and “results” section.

-    The authors reported “metabolic” alterations in some patients. How metabolomic analysis was performed?

Answer: the information was inserted in the “results” section of the text accordingly and the specific technique applied is described such as gas chromatography (GC), mass spectrometry (MS).

- Table 2: the Legend of this table is misleading. The authors claim that the table “summarizes” the “Results of investigation of LSS patients’ mtDNA in this study, but cite other groups work. I would suggest to clarify it. The authors should clearly indicate what is new and what has been already reported by others. In this regard: 1) more references should be included, and 2) in the manuscript should be specified that the mutations related to MT ATP-6 are found in “only” 10% of Leigh Syndrome patients. There is a very nice work from Marni J. Folk (MT-ATP6 mitochondrial disease variants: phenotypic and biochemical features analysis in 218 published cases and cohort of 14 new cases, Hum Mutat, 2019) that I would recommend to consider and discuss.

Answer: we have rewritten, and cleared which mutations were identified for the first time by us and by the other groups, as well as used the suggested paper to discuss and improve our manuscript. 

- Table 3: Same comment provided for Table 2.

Answer: we have rewritten, and cleared which mutations were identified for the first time by us and other groups. Here, we also used the suggested paper to discuss and improve our manuscript. 

- Figure 1: I would simplify the Figure by including only one eukaryotic cell, using two different arrows to indicate the nuclear and the mitochondrial compartment.

Answer: we made the change according to the suggestion.

-  Line 224: To which study the authors are referring here? Please specify it.

Answer: we added the additional information to the text to make it clear. Tise, C. G.; Verscaj, C. P.; Mendelsohn, B. A.; Woods, J.; Lee, C. U.; Enns, G. M.; Stander, Z.; Hall, P. L.; Cowan, T. M.; CusmanoOzog, K. P.  MTATP6  Mitochondrial Disease Identified by Newborn Screening Reveals a Distinct Biochemical Phenotype. Am. J. Med. Genet. A. 2023, 191 (6), 1492–1501. https://doi.org/10.1002/ajmg.a.63159.

-  Did the authors measured/recorded any metabolic dysfunction and/or immunodeficiencies associated with their mitochondrial variants? In the “Discussion” it seems that this is the case, but the information should be included in the “Result” section.

Answer: it was not possible to perform a complete metabolic study in all patients. The available data was inserted in the “results” section.

Comments on the Quality of English Language: Minor English editing is required.

Answer: we took special attention to this topic and refined the original version.
